# Neuron–Microglia Contact-Dependent Mechanisms Attenuate Methamphetamine-Induced Microglia Reactivity and Enhance Neuronal Plasticity

**DOI:** 10.3390/cells11030355

**Published:** 2022-01-21

**Authors:** Joana Bravo, Inês Ribeiro, Ana Filipa Terceiro, Elva B. Andrade, Camila Cabral Portugal, Igor M. Lopes, Maria M. Azevedo, Mafalda Sousa, Cátia D. F. Lopes, Andrea C. Lobo, Teresa Canedo, João Bettencourt Relvas, Teresa Summavielle

**Affiliations:** 1Addiction Biology, i3S-Instituto de Investigação e Inovação em Saúde, Universidade do Porto (UP), 4200-135 Porto, Portugal; joana.bravo@ibmc.up.pt (J.B.); afterceiro@ibmc.up.pt (A.F.T.); andrea.lobo@ibmc.up.pt (A.C.L.); teresa.canedo@ibmc.up.pt (T.C.); 2IBMC—Instituto de Biologia Molecular e Celular, Universidade do Porto (UP), 4200-135 Porto, Portugal; elva.andrade@ibmc.up.pt (E.B.A.); camila.portugal@ibmc.up.pt (C.C.P.); igor.lopes@ibmc.up.pt (I.M.L.); maria.azevedo@ibmc.up.pt (M.M.A.); mafsousa@ibmc.up.pt (M.S.); jrelvas@ibmc.up.pt (J.B.R.); 3ICBAS—Instituto de Ciências Biomédicas de Abel Salazar, Universidade do Porto (UP), 4050-313 Porto, Portugal; 4Escola Superior de Saúde, Politécnico do Porto, 4200-072 Porto, Portugal; 5Life and Health Sciences Research Institute (ICVS), School of Medicine, University of Minho, 4710-057 Braga, Portugal; id9531@alunos.uminho.pt; 6ICVS/3B’s—PT Government Associate Laboratory, 4806-909 Braga/Guimarães, Portugal; 7Immunobiology, i3S-Instituto de Investigação e Inovação em Saúde, Universidade do Porto (UP), 4200-135 Porto, Portugal; 8Glial Cell Biology, i3S-Instituto de Investigação e Inovação em Saúde, Universidade do Porto (UP), 4200-135 Porto, Portugal; 9Laboratory Animal Science, i3S-Instituto de Investigação e Inovação em Saúde, Universidade do Porto (UP), 4200-135 Porto, Portugal; 10Advanced Light Microscopy Scientific Platform, i3S-Instituto de Investigação e Inovação em Saúde, Universidade do Porto (UP), 4200-135 Porto, Portugal; 11Molecular Bionics Group—Institute for Bioengineering of Catalonia (IBEC), 08028 Barcelona, Spain; clopes@ibecbarcelona.eu; 12FMUP—Faculdade de Medicina da Universidade do Porto, 4200-319 Porto, Portugal

**Keywords:** methamphetamine, neuron-to-microglia, neuroprotection, contact-dependent, CD200, PSD95

## Abstract

Exposure to methamphetamine (Meth) has been classically associated with damage to neuronal terminals. However, it is now becoming clear that addiction may also result from the interplay between glial cells and neurons. Recently, we demonstrated that binge Meth administration promotes microgliosis and microglia pro-inflammation via astrocytic glutamate release in a TNF/IP_3_R2-Ca^2+^-dependent manner. Here, we investigated the contribution of neuronal cells to this process. As the crosstalk between microglia and neurons may occur by contact-dependent and/or contact-independent mechanisms, we developed co-cultures of primary neurons and microglia in microfluidic devices to investigate how their interaction affects Meth-induced microglia activation. Our results show that neurons exposed to Meth do not activate microglia in a cell-autonomous way but require astrocyte mediation. Importantly, we found that neurons can partially prevent Meth-induced microglia activation via astrocytes, which seems to be achieved by increasing arginase 1 expression and strengthening the CD200/CD200r pathway. We also observed an increase in synaptic individual area, as determined by co-localization of pre- and post-synaptic markers. The present study provides evidence that contact-dependent mechanisms between neurons and microglia can attenuate pro-inflammatory events such as Meth-induced microglia activation.

## 1. Introduction

Methamphetamine (Meth) is a potent and widely abused synthetic stimulant, classically recognized by its addictive potential [1]. Clinically, Meth use is associated with important neurological changes that result in impaired judgment, psychomotor agitation, aggression, psychosis, altered cognitive function, anxiety, and depression as well as other negative outcomes, such as cardiac dysfunction, dysregulation of body temperature, renal and liver failure, and higher risk of stroke (reviewed in [2]). At the cellular and molecular level, numerous publications have reported that Meth exposure targets the dopaminergic system, binding to the dopamine transporter (DAT) and leading to marked depletion of dopamine (DA), mainly in the striatum, but also in other regions that receive dopaminergic innervation, such as the hippocampus, amygdala, and frontal cortex [2,3]. Through its interaction with the vesicular monoamine transporter 2 (VMAT-2), Meth is also recognized to cause high oxidative stress and mitochondrial dysfunction [3]. In addition, Meth administration increases extracellular glutamate levels in the striatum and hippocampus, which seems to synergize with DA to cause methamphetamine-induced neurotoxicity [4]. The hippocampus is particularly affected by high glutamate release and by Meth administration, which results in impaired memory function [5,6]. Of note, antagonists of glutamate receptors were shown to block Meth-associated toxicity and limit its negative behavioral impact [7].

Importantly, it has become increasingly accepted that the interplay between neuronal and glial cells is also a relevant mechanism for Meth action [8,9]. Using human cerebral organoids, it was recently demonstrated at single-cell resolution that, upon exposure to Meth, most upregulated pathways were associated with immune response and oxidative stress [10]. In accordance, a single dose of Meth given to rats was sufficient to increase the levels of pro-inflammatory cytokines in several brain regions [11].

Recently, we have shown that a binge administration of Meth leads to microgliosis and microglia activation in a process mediated by astrocytes via glutamate release in a TNF/IP_3_R2-Ca^2+^-dependent manner [12]. In addition, we have also shown increased levels of pro-inflammatory cytokines IL-1ß and TNF in the striatum and hippocampus [12]. However, how Meth-induced glial reactivity may affect neurons is yet unknown.

The neuronal ability to sense the surrounding environment is crucial for maintaining homeostasis in the brain parenchyma [13]. There is now considerable evidence showing that immune and neuronal systems communicate through contact-dependent regulatory molecules or other soluble factors, such as neurotransmitters, neuromodulators, cytokines, neuropeptides, and miRNAs [13,14,15]. In this context, the communication between neurons and microglia seems to critically regulate neuroinflammatory responses [14,16,17]. In fact, it has been previously shown, in different neurological disorders, that neurons might attenuate microglia reactivity resorting to various neuroprotective mechanisms [17,18].

Several neuroimmune regulatory proteins, highly expressed in neurons, were already shown to modulate microglial activation, including CD200, CD22, CX3CL1, and CD95, among others [13]. Notably, receptors to these ligands are mostly restricted to myeloid cells. In particular, the interaction of CD200 and its receptor (CD200r) seems to act as a potent immune suppressor, promoting microglia quiescence [13]. In accordance, CD200-deficient mice display severe progression of neuroinflammation [19]. Opposing this, triggering the CD200/CD200r signaling pathway can limit microglia activation and inflammatory responses by inhibiting NF-κB and mitogen-activated protein kinase (MAPK) pathways, thus limiting the production of the inflammatory cytokines IL-1ß and IL-6 [18]. Also, CD200/CD200r signaling was shown to be involved in the spontaneous recovery of synaptic plasticity after stroke via the inhibition of microglia activation and reduction of inflammatory factors release [18]. Corroborating this, in animal models of Alzheimer’s disease, restoration of neuronal CD200 improved cognitive function and prevented further synaptic impairment [20]. 

In this scenario, understanding how glial activation by Meth affects the crosstalk between glial cells and neurons will provide a more comprehensive understanding of the mechanisms involved in modulation of psychostimulant-induced neuroinflammation. Here, we used primary neuronal cultures to investigate how neurons affect microglia activation under Meth exposure. Our results reveal that contact-dependent mechanisms between neurons and microglia, namely through CD200/CD200r signaling, attenuate Meth-induced microglia reactivity and enhance synaptic plasticity. 

## 2. Materials and Methods

### 2.1. Animals

Timely pregnant Wistar dams were necessary to obtain both 18-day embryos (E18) and postnatal day-one or -two pups (P1–P2). All procedures were conducted following the Directive 2010/63/EU and approved by the competent authorities, Direcção Geral de Alimentação e Veterinária (DGAV) and i3S Animal Ethical Committee (ref.2018-13-TS and DGAV 003891/2019-02-15). Researchers involved in animal experimentation were FELASA certified. All efforts were made to minimize animal suffering and the number of dams/pups used. Dams were housed under specific pathogen-free conditions and controlled environment (20 °C, 45–55% humidity) with free access to food and water.

### 2.2. Neuronal Cell Culture

Hippocampal, striatal, and mesencephalic neurons were obtained from Wistar rat embryos (E18) as described before [21]. Briefly, rat embryo brains were dissected in Hank’s buffer solution (HBSS) (Gibco, ThermoFisher Scientific, Waltham, MA, USA) and enzymatically treated with trypsin (1.5 mg/mL) (Gibco, ThermoFisher Scientific, Waltham, MA, USA) for 15 min [21]. The tissue was dissociated in serum-free Neurobasal medium (Gibco, ThermoFisher Scientific, Waltham, MA, USA) supplemented with B27 (Gibco, ThermoFisher Scientific, Waltham, MA, USA), glutamine (0.5 mM, Sigma-Aldrich, St. Louis, MO, USA), and gentamycin (0.12 mg/mL) (Gibco, ThermoFisher Scientific, Waltham, MA, USA) [21]. The cells were seeded on cell culture plates coated with 0.1 mg/mL of poly-d-lysine (PDL) (Sigma-Aldrich, St. Louis, MO, USA). Hippocampal cell cultures were supplemented with glutamate (25 μM, Sigma-Aldrich, St. Louis, MO, USA). Hippocampal cells were cultured at a density of 0.75 × 10^5^/cm^2^ and co-cultured striatal and mesencephalic neurons were plated at a density of 1.25 × 10^5^/cm^2^ (in a proportion of 2:1 striatal to mesencephalic cells). Cell cultures were kept at 37 °C in a humidified incubator with 5% CO_2_/95% air for 12 days in vitro (DIV). At DIV8, the medium was partially replaced. 

### 2.3. Microglia and Astrocyte Purified Cultures Obtained from Mixed Glial Cell Cultures

The mixed glial cell cultures (MGC) were obtained from newborn Wistar pups (P1–P2) [12,22]. The pups’ brains were dissected in HBSS with 1% Penicillin-Streptomycin (P/S) (Gibco, ThermoFisher Scientific, Waltham, MA, USA). The meninges were removed, and the cortex and hippocampi were collected. After complete tissue dissociation, DNAse I (0.1 U/mL) (Irvine, CA, USA) and 0.25% trypsin (Gibco, ThermoFisher Scientific, Waltham, MA, USA) were added and incubated for 15 min at 37 °C. The cells were suspended in Dulbecco’s Modified Eagle Medium (DMEM) (1x)/Glutamax (Gibco, ThermoFisher Scientific, Waltham, MA, USA) supplemented with 10% of FBS and 1% of P/S. The cell suspension was distributed by T-75 flasks coated with 0.01 mg/mL of PDL at a density of two brains/flask. The cell cultures were maintained for 10 days at 37 °C in a humidified incubator with 5% CO_2_/95% air; the medium was partially replaced at DIV4 and then totally replaced every 2 days. 

To prepare the purified microglial cultures, MGCs were shaken at 200 rpm for 2 h in an iNFORS HT Minitron incubator with a radius of 2.5 cm at 37 °C. The cell suspension obtained was centrifuged at 1200 rpm for 10 min and suspended in Dulbecco’s Modified Eagle Medium: Nutrient Mixture F-12 (DMEM/F-12) (Gibco, ThermoFisher Scientific, Waltham, MA, USA) supplemented with 10% of FBS and 1% of P/S. Purified microglia were plated at a density of 0.5 × 10^5^/cm^2^. Purity of microglia cell cultures was 99% as previously reported [23].

To obtain purified astrocytic cultures, the MGCs were shaken at 220 rpm overnight to remove non-astrocytic cells, and adherent cells were detached with 0.25% trypsin. The cell suspension was centrifuged at 1200 rpm for 10 min and partially split in DMEM/Glutamax supplemented with 10% of FBS and 1% of P/S in a non-coated flask. 

Purified astrocytes were obtained after two consecutive splits. Cells were plated for the experiment at a density of 0.5 × 10^5^/cm^2^.

### 2.4. Microglia Incubation with Neuron-Conditioned Medium

At DIV12, the neuronal cell cultures were treated with Meth 100 μM (Sigma-Aldrich, St. Louis, MO, USA) for 24 h [12,24]. Untreated neuron-conditioned medium (NCM Ctrl) and conditioned medium from Meth-treated neurons (NCM Meth) were then collected, centrifuged (1200 rpm, 1 min), and added to microglial cultures. 

### 2.5. Co-Cultures of Neurons and Microglia in Microfluidic Devices 

Co-cultures of neurons and microglia were performed in an Axon Investigation System, adapted from [25] (AXIS150, Merck Millipore, Burlington, MA, USA). The microfluidic devices were placed on the top of a glass coverslip (22 mm × 22 mm) coated with 0.1 mg/mL PDL, channel side down, creating a microfluidic chamber composed of two compartments separated by 150 μm length × 5 μm height × 10 μm width microgrooves. Neuronal cells were plated in one of the compartments at a density of 1 × 10^5^/device. At DIV12 for neurons, microglial cells were plated in the opposite chamber at a density of 1 × 10^5^/device. 

### 2.6. Astrocytic-Conditioned Medium

At DIV2, purified astrocytes were incubated with Meth 100 µM for 24 h. The medium from untreated astrocytes (ACM Ctrl) or from Meth-treated astrocytes (ACM Meth) was collected and centrifuged at 1200 rpm for 1 min. 

### 2.7. Treatment of Neuron–Microglia Co-Cultures

To carry out this study, three different conditions were used: (1) Meth 100 µM was added to the neuronal side in the microfluidic device for 24 h; (2) the conditioned media obtained from astrocytes treated with Meth 100 µM for 24 h (ACM Meth) or from control astrocyte cultures (ACM Ctrl) were added to the microglial/axonal compartment for 24 h; (3) ACM Ctrl or ACM Meth were added to the microglia/axonal compartment, and Meth 100 µM was added to the neuronal compartment for 24 h.

### 2.8. Immunocytochemistry and Image Acquisition

Cells were fixed with 4% PFA, permeabilized with 0.25% Triton X-100, and blocked with 3% BSA. Next, cells were incubated overnight at 4 °C with the primary antibody, as per the manufacturer’s recommendations, washed, and incubated with the secondary antibody for 1 h at RT. Then, cells were incubated with DAPI for 5 min or Hoechst 33,342 (1 µg/mL) for 10 min for the neuronal cell viability assay. Coverslips were mounted with fluorescent mounting media (Dako, Agilent, Santa Clara, CA, USA). Imaging was performed using a Zeiss AxioImager Z1 fluorescence microscope equipped with an Axiocam MR v3.0 camera, an HXP 120 light source, and a 40x/1.30 objective. The following filter sets were used to image DAPI, Alexa 488, Alexa 568, and DyLight 650, respectively: 49—Excitation: G 365 Beam Splitter: FT 395 Emission: BP 445/50; 38HE—Excitation: BP 470/40 (HE) Beam Splitter: FT 495 (HE) Emission: BP 525/50 (HE); 43HE—Excitation: BP 550/25 (HE) Beam Splitter: FT 570 (HE) Emission: BP 605/70 (HE); 50—Excitation: BP 640/30 Beam Splitter: FT 660 Emission: BP 690/50. Antibodies are detailed in Appendix A. 

### 2.9. Phagocytic Assay

The phagocytic assay was performed as previously described [12,23]. Microglial cultures were incubated with fluorescent latex beads 0.5 μm in diameter (Sigma-Aldrich, St. Louis, MO, USA) for 30 min at 37 °C. Briefly, the beads were diluted in cell culture medium (1:1000); then, cells were washed then fixed with 4% PFA, and the staining for iba1 (Wako) was performed. The number of beads per cell was counted. 

### 2.10. Fluorescence Intensity Quantification

The fluorescence intensity of microglial cells was quantified using the ImageJ software [26], as previously reported [12]. Fluorescence channels were separated, the image converted to 32-bit, and the background subtracted using 50%-off pixels radius by the rollerball algorithm. Then, cells were segmented using the Triangle algorithm with automatic complementation for both the bottom and upper threshold ramps, with background values set to black and foreground values set to red. Individual segmented cells were transposed to ImageJ’s ROI manager using the “analyze particles” tool. A range between 1000 and infinity of non-calibrated pixels defined each segmented cell. Mean gray values for the intensities were returned for each cell individually using the multi-measure function on ImageJ’s ROI manager.

### 2.11. RNA Extraction, cDNA Synthesis, and qRT-PCR

RNA from cell cultures was isolated using the PureLink^®^RNA Mini-Kit (ThermoFisher Scientific, Waltham, MA, USA) according to the manufacturer’s specifications, and its quality was checked by the Experion automated electrophoresis system (Bio-Rad, CA, USA) [12]. Of note, in the microglial and neuronal cell cultures, the microfluidic device was removed, and the RNA was extracted from total cell culture.

The synthesis of cDNA was performed using 500 ng of RNA through SuperScript^®^ III First-Strand Synthesis SuperMix (Invitrogen, Waltham, CA, USA). The qRT-PCR reactions, using equal amounts of total RNA from each sample, were performed on the CFX96 Touch^TM^ Real-Time PCR Detection System (Bio-Rad), using the iTaq™ universal SYBR^®^ Green supermix (Bio-Rad). All primers (Sigma-Aldrich, St. Louis, MO, USA) are described in Appendix A. Raw data were analyzed using the 2^−∆CT^ method, with S18 serving as the internal control.

### 2.12. Flow Cytometry of Microglia 

Cells were collected from microfluidic devices using Accutase (Gibco, ThermoFisher Scientific, Waltham, MA, USA) for 5 min, centrifuged at 1200 rpm for 5 min, washed with PBS 1x, and incubated with Zombie Aqua (BioLegend, CA, USA) for dead cell exclusion. After washing, cells were fixed with 2% PFA, resuspended in FACS buffer (2% BSA in PBS), and incubated with the antibodies described in Appendix A for 20 min at 4 °C (adapted from [12,27]. Data were acquired in a FACSCanto II flow cytometer (BD Biosciences, Franklin Lakes, NJ, USA). Post-acquisition analysis was performed using FlowJo software v10 (Tree Star, Ashland, OR, USA). 

### 2.13. Quantification of Synaptic Proteins

The acquired images were analyzed using ImageJ, as described elsewhere [28,29,30]. The puncta were analyzed for number, area, and intensity in the selected region, per dendritic length. Briefly, neurite isolated segments containing both PSD95 and VGlut1 stainings were thresholded using Li and Moments algorithms, respectively. The obtained mask was applied on the original images to determine the fluorescence intensity and area. The background value was then subtracted from the obtained intensity value, and this result was multiplied by the puncta area, obtaining the intensity per area. The thresholded signals were then used to determine the co-localization. For this, VGlut1 channel was set as binary, and PSD95 channel was used to define the total puncta. The puncta positive for VGlut1 and PSD95 were defined as co-localized.

### 2.14. Statistical Analysis

Data presentation is described in figure legends. Identification and removal of outliers were performed with the automatic GraphPad Prism^®^ software (version 9.1.2. for macOS) using the ROUT (robust non-linear regression) method with *Q* = 5%.

Statistical analyses for qRT-PCR and flow cytometry data were performed in the GraphPad Prism^®^ software using paired Student’s *t*-test. For immunocytochemistry data, a linear mixed model analysis was performed using SAS^®^ Visual Statistics with REML function, followed by Tukey–Kramer multiple comparison test, using the culture as a random variable. The data of PSD95 positive for VGlut1 intensity do not follow the normal distribution, so a transformation of the data was carried out using the square root function before the linear mixed model application. Adjusted *p*-value was considered for these data. Differences were considered at the significance level of *p* < 0.05.

## 3. Results

### 3.1. Neurons Exposed to Meth Do Not Promote Microglial Activation through Contact-Dependent or -Independent Mechanisms

We have previously shown that the conditioned media of astrocytes (ACM) exposed to Meth can activate microglial cells [12]. Thus, we started by evaluating the role of neurons in the activation of microglia under the same conditions of Meth exposure. To this end, we incubated primary microglia with conditioned media obtained from primary hippocampal cultures treated with Meth (NCM Meth) or with conditioned media obtained from control hippocampal cultures (NCM Ctrl) (Figure 1A) for 24 h. We found that NCM Meth did not promote increased expression of classical microglial activation markers, such as iNOS (Figure 1B) and iba 1 (Figure 1C). The phagocytic capacity of microglia exposed to NCM Meth was found to be decreased (*F* (1540) = 4.35, *p* < 0.05) (Figure 1D). Furthermore, we observed no differences in the mRNA levels of pro-inflammatory cytokines TNF, IL-1β, and IL-6 (Figure 1E) or in the mRNA levels of anti-inflammatory cytokines IL-10 and TGF-β (Figure 1F). We also verified that the dose of Meth used in the neuronal cultures was not affecting cell viability (Appendix A). 

To verify whether these results were specific for hippocampal neurons, which do not express some of the classical targets of Meth, such as the dopamine transporter (DAT) and the vesicular monoamine transporter (VMTA2), we repeated this evaluation using a co-culture of striatal GABAergic neurons and dopaminergic neurons from the mesencephalic region (Appendix A). Remarkably, we observed that NCM Meth from these co-cultures also did not activate microglial cells, as it did not result in increased iNOS expression (Appendix A), iba 1 intensity, or changes in phagocytic activity (Appendix A). These results suggest that, upon Meth exposure, neurons do not induce microglial activation through contact-independent mechanisms. 

Since the conditioned media of neurons exposed to Meth did not induce microglia activation, and taking into consideration that microglia and neurons also communicate through cell-to-cell contact via several regulatory molecules [13,14], we tested the hypothesis that the crosstalk between neurons and microglia under Meth exposure could be contact-dependent. We used co-cultures of microglia and hippocampal neurons in microfluidic devices, which allow direct contact between microglia and axons, and exposed neurons to Meth for 24 h (Figure 2A). Of note, hippocampal neurons can extend long axons throughout the grooves that separate the two microfluidic compartments, while striatal neurons cannot. Our findings revealed that neurons exposed to Meth did not significantly increase iNOS expression (Figure 2B) or microglia phagocytic capacity (Figure 2D) and only modestly increased iba1 expression (Figure 2C). Since arginase 1 can outcompete iNOS to downregulate production of nitric oxide and is, therefore, considered a good anti-inflammatory marker for microglia [31], we also evaluated the expression of arginase 1 in these co-cultures. However, we found that in the presence of Meth, arginase 1 was decreased (*F* (1892) = 24.71, *p* < 0.0001) (Figure 2E). This set of results suggests that exposing neurons to Meth was not sufficient to promote a robust pro-inflammatory state in microglia. To further confirm this, the mRNA expression levels of the pro-inflammatory cytokines TNF-α, IL-1β, and IL-6 and the anti-inflammatory cytokines Il-10 and TGF-β were evaluated. No differences were found between groups (Figure 2F,G). Together, these data indicate that neurons exposed to Meth do not trigger pro-inflammation in microglia in a cell-autonomous way, either through contact-dependent or -independent mechanisms.

### 3.2. Microglia Activation Induced by Meth-Exposed Astrocytes Is Partially Prevented by Neurons 

Since we have recently shown that Meth-induced microglial reactivity requires both astrocyte-released TNF and glutamate [12], we next sought to verify whether this was also the case in co-cultures of neurons and microglia. To do so, co-cultures of neurons and microglia were performed in microfluidic devices, and conditioned media of primary astrocytes, treated with Meth (ACM Meth) or untreated (ACM Ctrl), were added to the microglial/axonal compartment (Figure 3A). Confirming our previous results [12], ACM Meth increased iNOS levels in microglia (*F* (1729) = 120.67, *p* < 0.0001), independently of the presence of primary neurons (Figure 3B). However, no significant effects were observed in iba 1 expression (Figure 3C) or microglia phagocytic capacity (Figure 3D). Of note, arginase 1 was concomitantly increased in ACM Meth-treated microglia (*F* (1591) = 59.30, *p* < 0.0001) (Figure 3E). Because we previously reported that ACM Meth increased the transcript expression of pro-inflammatory cytokines in microglia cultures [12], we also evaluated the mRNA levels of the pro-inflammatory cytokines TNF, IL-1β, and IL-6. Interestingly, no significant changes were observed (Figure 3F). The mRNA expression levels of the anti-inflammatory cytokines IL-10 and TGF-β were also unaltered (Figure 3G). Next, we repeated these experiments, adding Meth to the neuronal compartment while still exposing microglia to ACM Meth (Appendix A). In these conditions, we also observed an increase in both iNOS (*F* (1942) = 53.45, *p* < 0.0001) and arginase 1 expression (*F* (1942) = 53.45, *p* < 0.0001) (Appendix A), while the phagocytic capacity (Appendix A) and the cytokines’ mRNA expression levels were not affected (Appendix A). Importantly, if we compare the present results with those that we have previously reported when exposing microglia to ACM [12], we no longer observe increased IL-1β and IL-6, and, although iNOS expression is still augmented, its competitor arginase 1 is also higher. Collectively, these results suggest that Meth-induced microglia activation via astrocytes is attenuated in the presence of neuronal cells.

### 3.3. Neurons Increased Self-Protection from Meth-Induced Activation of Microglia through Contact-Dependent Mechanisms

Because neurons can use several contact-dependent mechanisms to protect themselves from reactive microglia [14,16], we next evaluated the expression of neuroimmune regulatory molecules known to downregulate the microglial pro-inflammatory response. To do so, neurons and microglia were co-cultured in microfluidic devices, and ACM Meth or ACM Ctrl was added to the axonal/microglial side. After 24 h, the mRNA levels of ligand–receptor pairs relevant to the crosstalk between neurons and microglia were determined in the axonal/microglial compartment. We found a significant increase in CD200 mRNA levels (*p* < 0.05) without significant changes in its receptor CD200r (Figure 4A). The expression of CD22 and its receptor CD45 was not altered (Figure 4B). We also observed an increase in the mRNA levels of CX3CL1 that was close to reaching significance (*p* = 0.0565) but no differences for its receptor CX3CR1 (Figure 4C). Concerning the pair CD95–CD95L, the mRNA expression levels of CD95 were not altered (Figure 4D), and CD95L mRNA levels were not detectable. To further confirm these results, we also evaluated the protein expression of CD95 and CD200r specifically in microglia by flow cytometry, and no differences were observed either in CD95 (Figure 4E) or in CD200r levels (Figure 4F). These results indicate that neurons in contact with microglia activated via Meth-exposed astrocytes increased the expression of CD200, which is a well described self-protection mechanism.

### 3.4. Synaptic Proteins Expression Increases in Co-Cultures of Neurons and ACM Meth-Microglia

Since we found increased CD200 levels and the signaling of CD200/CD200r has been associated with the modulation of microglia activation and synaptic plasticity in other neurological diseases [18,20], we evaluated how exposure to ACM Meth was affecting synaptic proteins. For that, we used the same co-cultures of neurons and microglia in microfluidic devices with ACM Meth or ACM Ctrl added to the axonal/microglial side and evaluated the expression and co-localization of PSD95 and VGlut1. PSD95 is a post-synaptic marker and the most abundant scaffold protein of excitatory synapses, while VGlut1 is a presynaptic maker, highly expressed in the mouse hippocampus, specifically in excitatory synapses. Both proteins display a punctate distribution, and their co-localization is defined as a synapse [29,30]. Compared with the ACM Ctrl condition, when ACM Meth was added to the axonal/microglial compartment, we observed an increase in PSD95 puncta number (*F* (1254) = 11.92, *p* < 0.001), area (*F* (1252) = 29.38, *p* < 0.0001), and intensity (*F* (1251) = 51.97, *p* < 0.0001) (Figure 5A–D) and a decrease in VGlut1 puncta numbers (*F* (1258) = 5.71, *p* < 0.05) (Figure 5E,F). However, no significant differences were observed in VGlut1 puncta area or intensity (Figure 5E–H). As the co-localization between both markers is accepted as an indicator of active synapses (since both pre- and postsynaptic terminals are present) [30], we also evaluated their co-localization and found a significant increase in the PSD95/VGlut1-positive puncta area (*F* (1247) = 17.53, *p* < 0.0001) and intensity (*F* (1244) = 49, adj *p* < 0.0001) (Figure 5I–L), reflecting an increase in synaptic area in the presence of ACM Meth, but no significant differences in PSD95/VGlut1-positive puncta numbers (Figure 5I,J). Collectively, these data show that neuronal/microglial co-cultures present higher levels of synaptic proteins and that synapses display an increased area of PSD95 when exposed to the ACM of Meth-treated astrocytes. 

## 4. Discussion

We have recently demonstrated that an acute binge Meth exposure (4 × 5 mg/kg with 2 h intervals) requires astrocytes to induce microglia activation through glutamate release in a TNF/IP_3_R2-Ca^2+^-dependent manner [12]. Herein, we explored in vitro how neurons may contribute to modulate microglia under Meth, demonstrating that contact-dependent mechanisms between neurons and microglia seem to attenuate Meth-induced microglia activation. 

Multiple mechanisms mediating Meth-induced neurotoxicity have been widely described, such as impairment of the dopaminergic system, increase of oxidative stress, mitochondrial dysfunction, and glutamate excitotoxicity [2,32,33]. The primary goal of this study was to further identify other mechanisms involved in Meth-mediated inflammation that could possibly rely on neuron–glia communication. As the crosstalk between microglia and neurons can occur by contact-dependent and/or -independent mechanisms, we first investigated whether contact-independent mechanisms were sufficient to mediate microglia activation by Meth-exposed neurons. Using a strategy similar to the one we have previously used to explore the contribution of astrocytes in Meth-induced microglia activation [12], we verified that neuron-conditioned media obtained from neurons exposed to Meth failed to induce a pro-inflammatory profile in primary microglia. Since these results could be explained by the fact that hippocampal neurons lack the Meth-binding proteins DAT and VMAT-2, we used co-cultures of striatal and dopaminergic neurons to confirm that this was not the case. A second possibility was that direct contact between neurons and microglia could be necessary to promote microglia activation, which we explored by resorting to neuron–microglia co-cultures in microfluidic chambers. Interestingly, although arginase outcompetes iNOS and limits NO production [34], it did not lead to upregulation of iNOS, and we only found a modest increase in iba 1 and decreased arginase 1 expression in microglial cells. These results clarified that neither direct nor indirect neuron–microglia contact was sufficient to promote a robust inflammatory response under Meth exposure, indicating that astrocyte mediation was, in fact, necessary, as we previously reported [12]. Confirming this, when we added ACM Meth to the axonal/microglial compartment, microglia exhibited increased iNOS expression. Nevertheless, if we compare these results with those obtained in the absence of neurons, we verify that this response is clearly attenuated, indicating that neurons may be promoting this attenuation.

Previous studies demonstrated that neurons may induce an anti-inflammatory profile in microglial cells to protect themselves from excessive microglia activation [14,16]. Several neuronal membrane proteins, including CD200, CD22, CD95, CD47, and other proteins such as NCAM or ICAM-5, were shown to modulate microglial activation through specific counter-receptors [14]. As an example, neurons can contribute to inhibit the lipopolysaccharide (LPS)-induced release of NO and pro-inflammatory cytokines by glial cells [35]. As such, understanding the role of these ligand–receptor pairs has gained increased relevance over the last few years. Among the most explored, CD200 is highly expressed in neurons and other cells, but its receptor, CD200r, is found predominantly in myeloid cells. Interestingly, knocking out CD200 in mice leads to robust activation of microglia [36], while overexpressing it seems to protect from different pro-inflammatory events [14]. Likewise, the interaction of CD22 and its microglial receptor CD45 was shown to prevent LPS-induced neuroinflammation [37]. It was also reported that neurons may induce microglial apoptosis through increased expression of CD95L (or Fas ligand) [38], while CD95L expression in hippocampal neurons protects them against reactive T cells [39]. In the present work, we found increased expression of CD200 when ACM Meth was added to the axonal/microglial compartment, which likely reflects a neuronal defense strategy against astrocyte-driven activation of microglia. Supporting this view, in the presence of neuronal cells, ACM Meth no longer promotes increased levels of IL-1ß and IL-6, which is compatible with the activation of the CD200/CD200r signaling pathway and NF-κB pathway inhibition [18]. Of note, there is still an important gap in the literature concerning which receptors may be mediating Meth action over inflammation-related pathways [40]. 

Interestingly, the CD200/CD200r signaling pathway was reported to increase dendritic spine density and PSD95 [18], an essential indicator of synaptic plasticity [41,42]. ACM Meth resulted in an increased number of PSD95 puncta, indicating higher density of PSD95 along the neurite [43], and increased area and intensity of PSD95 puncta co-localized with VGLUT1, which is a mechanism that can allow more anchoring of receptors, such as NMDA-R and AMPA-R, thus promoting increased synapse strength [30]. Of note, the dynamics of PSD95 seem to respond also to glutamate variations, which we have reported to be increased in ACM Meth [12]. On the other hand, we found decreased VGlut1 puncta numbers, a major player in vesicular glutamate uptake, in which dysregulation in response to Meth was previously reported [44]. Acute exposure to psychoactive substances, such as Meth, has long been associated with high levels of synaptic plasticity, contributing to the drug-driven effects in neuronal function [45,46]. Although the exact mechanisms regulating Meth-associated changes in synaptic plasticity and their direct link to CD200 require further exploration, there is already enough evidence supporting the involvement of CD200/CD200r signaling in suppressing inflammatory responses. Interestingly, this seems to be exerted through reduced activation of MAPK [18], which is also involved in the regulation of classic RhoGTPases, such as RhoA, Rac1, and Cdc42, that are well known for their roles in spine formation and maintenance [47,48].

## 5. Conclusions

In summary, using microfluidic chambers as an effective tool to explore contact-dependent mechanisms between neurons and microglia, we report that hippocampal neurons in direct contact with primary microglia counteract microglia activation by increasing the expression of the neuroimmune regulatory molecule CD200. Furthermore, our data also suggest that the activation of the CD200/CD200r signaling pathway is likely contributing to increased synaptic plasticity. These results highlight CD200 and its receptor as molecules with promising therapeutic potential in the field of addiction.

## Figures and Tables

**Figure 1 cells-11-00355-f001:**
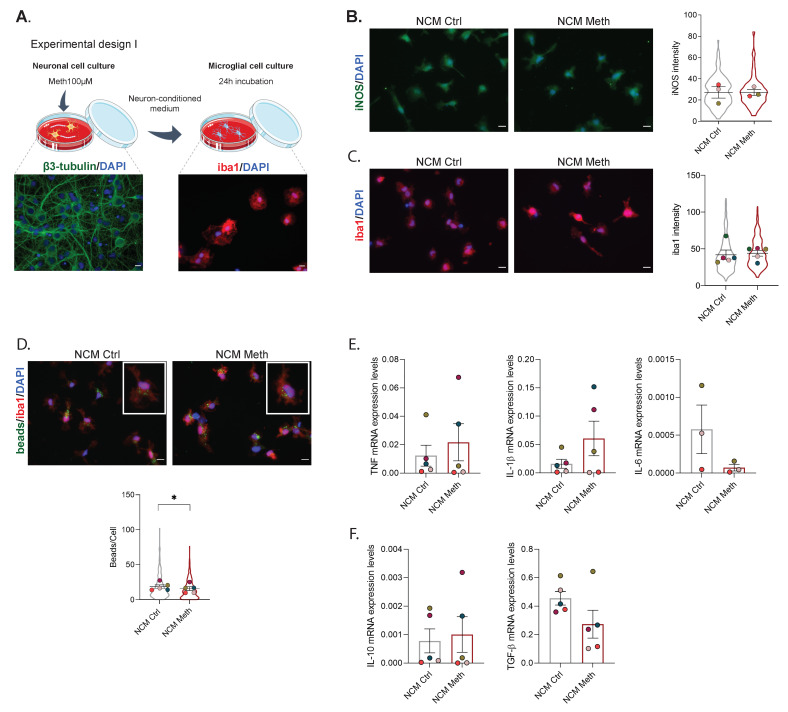
Neurons exposed to Meth do not promote microglial activation through contact-independent mechanisms. (**A**) Experimental design I—Neuronal cell cultures from the hippocampus were incubated with 100 µM Meth for 24 h. Following that, primary microglial cell cultures were incubated for 24 h with the conditioned media obtained from neurons treated with Meth (NCM Meth) or from control neuronal cell cultures (NCM Ctrl). (**B**) Fluorescence imaging of microglial cells immunolabeled for iNOS (green) incubated with NCM Ctrl or NCM Meth for 24 h. Results express the iNOS intensity (mean ± SEM) of three independent cultures (NCM Ctrl—132 cells; NCM Meth—143 cells). (**C**) Fluorescence imaging of microglial cells immunolabeled for iba1 (red) incubated with NCM Ctrl or NCM Meth for 24 h. Results express iba1 intensity (mean ± SEM) of five independent cultures (NCM Ctrl—252 cells; NCM Meth—265 cells). (**D**) Fluorescence imaging of microglial cells immunolabeled for iba1 (red) incubated with microbeads (green) and treated with NCM Ctrl or NCM Meth for 24 h. Results represent the number of beads per cell (mean ± SEM) of five independent cultures (NCM Ctrl—267 cells; NCM Meth—264 cells). Statistical analysis for (**B**–**D**) was performed using a linear mixed model followed by Tukey–Kramer comparison test (* *p* < 0.05). Symbol colors represent the mean of each independent cell culture and the violin plots the variability of all cells quantified. Scale bar: 10 µm. (**E**) RT-qPCR for TNF, Il-1β, and Il-6 from microglia exposed to NCM Ctrl and NCM Meth for 24 h. (**F**) RT-qPCR for Il-10 and TGFβ from microglia exposed to NCM Ctrl and NCM Meth for 24 h. In both cases, results were normalized to the S18 gene and are expressed as the mean ± SEM of five independent cultures. Statistical analysis was performed using paired Student’s *t*-test.

**Figure 2 cells-11-00355-f002:**
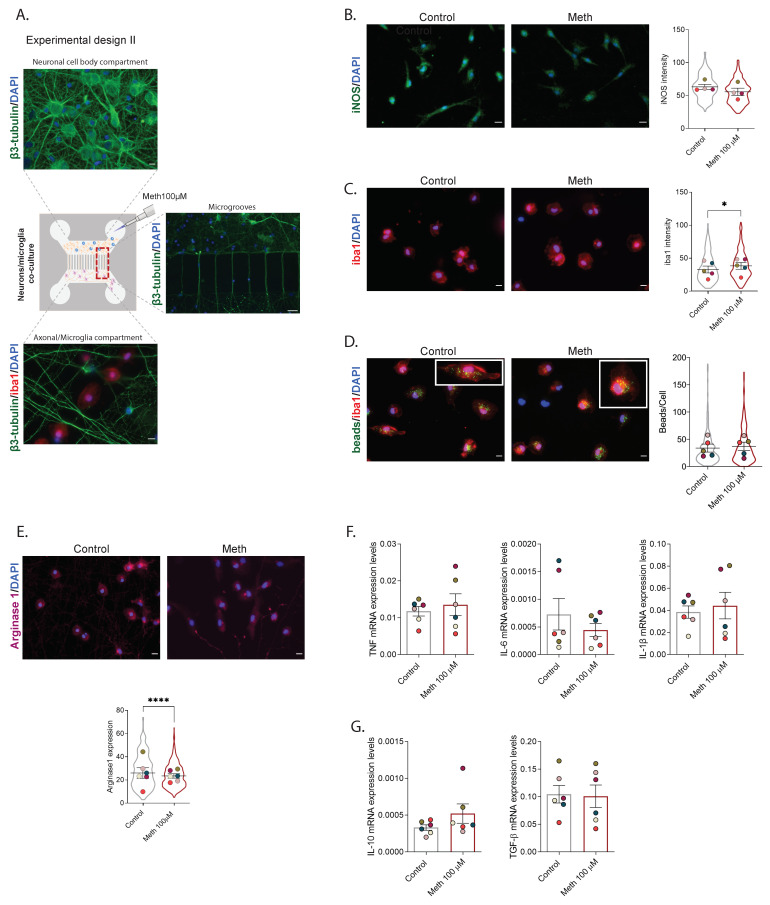
Neurons exposed to Meth and in direct contact with microglia do not promote its activation. (**A**) Experimental design II—neuronal cell culture from the hippocampus was seeded in one side of a microfluidic device, and the axons were allowed to extend to the other compartment for 12 days. On DIV12, microglial cells were seeded in the axonal compartment. On DIV13, Meth 100 µM was added to the neuronal compartment for 24 h. (**B**) Fluorescence imaging of microglial cells cultured in a microfluidic device with neurons exposed to Meth 100 µM for 24 h. Microglial cells were immunolabeled for iNOS (green), and the results express the iNOS intensity (mean ± SEM) of four independent cultures (Ctrl—297 cells; Meth—325 cells). (**C**) Fluorescence imaging of microglial cells immunolabeled for iba1 (red) incubated with NCM Ctrl or NCM Meth for 24 h. Results express iba1 intensity (average ± SEM) of five independent cultures (Ctrl—199 cells; Meth—206 cells)**.** (**D**) Fluorescence imaging of microglial cells cultured in a microfluidic device with neurons exposed to Meth 100 µM for 24 h. Microglia were incubated with microbeads (green) and immunolabeled for iba1 (red). Results represent the number of beads per cell (mean ± SEM) of five independent cultures (Ctrl—268 cells; Meth—211 cells). (**E**) Fluorescence imaging of microglial cells cultured in a microfluidic device with neurons exposed to Meth 100 µM for 24 h. Microglial cells were immunolabeled for arginase 1 (magenta), and results express the arginase 1 intensity (mean ± SEM) of six independent cultures (Ctrl—489 cells; Meth—410 cells). Statistical analysis for B, C, D, and E was performed using a linear mixed model followed by Tukey–Kramer comparison test (* *p* < 0.05 and **** *p* < 0.0001). Symbol colors represent the mean of each independent cell culture and the violin plots the variability of all cells quantified. Scale bar: 10 µm. (**F**) RT-qPCR for TNF, Il-1β, and Il-6 from microglia and neuron co-culture cells, cultured in a microfluidic device where neurons were exposed to Meth 100 µM for 24 h. (**G**) RT-qPCR for Il-10 and TGFβ from microglia and neuron co-cultured cells, in a microfluidic device where neurons were exposed to Meth 100 µM for 24 h. In both cases, results were normalized to the S18 gene and are expressed as the mean ± SEM of six independent cultures. Statistical analysis was performed using paired Student’s *t*-test.

**Figure 3 cells-11-00355-f003:**
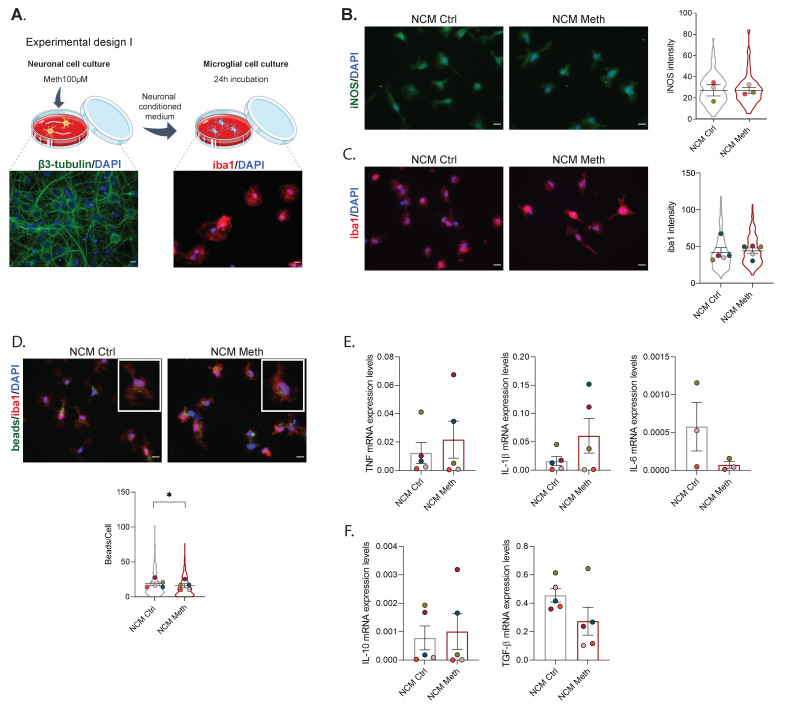
Neurons partially prevented microglia activation via Meth-exposed astrocytes. (**A**) Experimental design III—primary cell cultures of astrocytes were incubated with Meth 100 µM for 24 h. Neuronal cells from the hippocampus were seeded in one side of a microfluidic device, and the axons were allowed to extend to the other compartment for 12 days. On DIV12, microglial cells were seeded in the axonal compartment. The conditioned media obtained from astrocytes treated with Meth (ACM Meth) or from control astrocyte cultures (ACM Ctrl) were added to the microglia/axonal compartment for 24 h. (**B**) Fluorescence imaging of microglial cells co-cultured with neurons and exposed to the ACM Ctrl or ACM Meth for 24 h. Microglial cells were immunolabeled for iNOS (green), and results express the iNOS intensity (mean ± SEM) of five independent cultures (ACM Ctrl—405 cells; ACM Meth—329 cells). (**C**) Fluorescence imaging of microglial cells immunolabeled for iba1 (red) incubated with NCM Ctrl or NCM Meth for 24 h. Results express iba1 intensity (mean ± SEM) of four independent cultures (Ctrl—151 cells; Meth—184 cells). (**D**) Fluorescence imaging of microglial cells co-cultured with neurons and incubated with ACM Ctrl or ACM Meth for 24 h. Microglia were incubated with microbeads (green) and immunolabeled for iba1 (red). The results represent the number of beads per cell (mean ± SEM) of five independent cultures (ACM Ctrl—183 cells; ACM Meth—248 cells). (**E**) Fluorescence imaging of microglial cells co-cultured with neurons and exposed to ACM Ctrl or ACM Meth for 24 h. Microglial cells were immunolabeled for arginase 1 (magenta), and results express the arginase 1 intensity (mean ± SEM) of six independent cultures quantified (ACM Ctrl—310 cells; ACM Meth—287 cells). Statistical analysis for B, C, D, and E was performed using a linear mixed model followed by Tukey–Kramer comparison test (**** *p* < 0.0001). Symbol colors represent the mean of each independent cell culture and the violin plots the variability of all cells quantified. Scale bar: 10 µm. (**F**) RT-qPCR for TNF, Il-1β, and Il-6 from microglia and neuron co-culture cells, cultured in a microfluidic device where neurons were exposed to Meth 100 µM for 24 h. (**G**) RT-qPCR for Il-10 and TGFβ from microglia and neurons co-culture cells, cultured in a microfluidic device with microglia exposed to ACM Ctrl or ACM Meth for 24 h. In both cases, results were normalized to the S18 gene and are expressed as mean ± SEM of five independent cultures. Statistical analysis was performed using paired Student’s *t*-test.

**Figure 4 cells-11-00355-f004:**
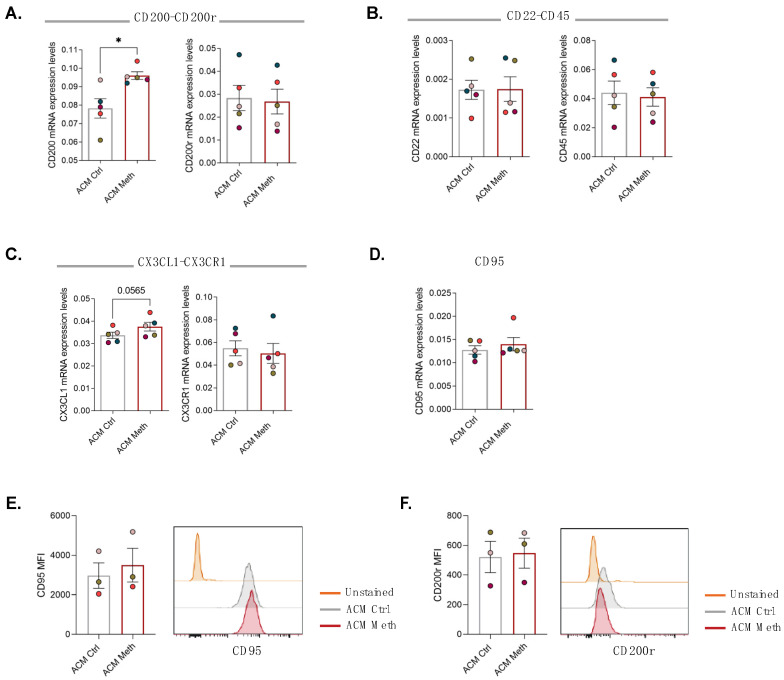
Neurons increased self-protection through contact-dependent mechanisms. RT-qPCR for ligand–receptor immunoregulatory pairs from microglia and neurons co-cultured in microfluidic devices, where microglial cells were exposed to ACM Ctrl or ACM Meth for 24 h. (**A**) Results for CD200 and CD200r, (**B**) CD22 and CD45, (**C**) CX3CL1 and CX3CR1, and (**D**) CD95. The results were normalized to the S18 gene and are expressed as the mean ± SEM of five independent cultures. Symbol colors represent the mean of each independent cell culture. Statistical analysis was performed using paired Student’s *t*-test (* *p* < 0.05). Quantification of (**E**) CD95 and (**F**) CD200r on microglia by flow cytometry, presented as median fluorescence intensity (MFI). Results are expressed as mean ± SEM of three independent cultures. Symbol colors represent the mean of each independent cell culture. Statistical analysis was performed using paired Student’s t-test. Representative histograms are shown; orange histogram represents unstained cells, gray the ACM Ctrl group, and red the ACM Meth group.

**Figure 5 cells-11-00355-f005:**
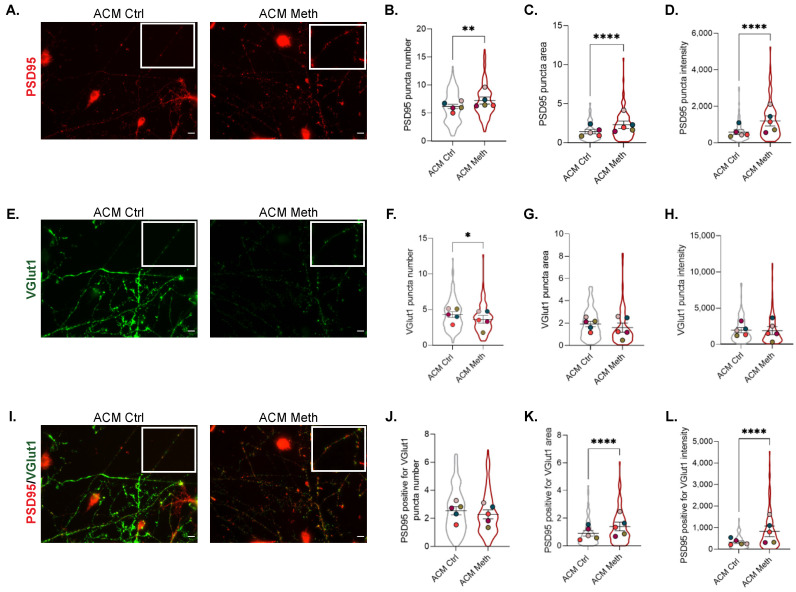
Synaptic plasticity was concomitantly increased. (**A**) Fluorescence imaging of microglial cells co-cultured with neurons and exposed to ACM Ctrl or ACM Meth for 24 h. Cells were immunolabeled for PSD95 (red), and results represent the PSD95 puncta (**B**) number, (**C**) area, and (**D**) intensity. (**E**) Fluorescence imaging of microglial cells co-cultured with neurons and exposed to ACM Ctrl or ACM Meth for 24 h. Microglial cells were immunolabeled for vGlut1 (green), and results represent the Vglut1 puncta (**F**) number, (**G**) area, and (**H**) intensity. (**I**) Fluorescence imaging of microglial cells co-cultured with neurons and exposed to ACM Ctrl or ACM Meth for 24 h. Cells were immunolabeled for PSD95 (red) and VGlut1 (green), and results represent the (**J**) PSD95 plus vGlut1-positive puncta number, (**K**) puncta area, and (**L**) puncta intensity. Symbol colors of each graph represent the mean of each independent cell culture (*n* = 5) and the violin plots the variability of all neuronal segments quantified (ACM Ctrl—149; ACM Meth—111 segments). Statistical analysis was performed using a linear mixed model followed by Tukey–Kramer comparison test. Scale bar: 10 µm. * *p* < 0.05, ** *p* < 0.01, and **** *p* < 0.0001.

## Data Availability

Not applicable.

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
