# Peer review of "Neuron–Microglia Contact-Dependent Mechanisms Attenuate Methamphetamine-Induced Microglia Reactivity and Enhance Neuronal Plasticity"

_cells, 2022, doi:10.3390/cells11030355_

Round 1

Reviewer 1 Report

Comment to Authors: Cells-1530608

In this manuscript, Bravo et al report their findings on the contact-dependent mechanism between neurons and microglia in which neurons can partially prevent Meth-induced microglia activation via astrocytes. Experiments were well designed and the manuscript, in general, is well written. However, there are a few minor details that should be corrected and some language mistakes need to be carefully checked.

Specific comments on the manuscript

Introduction

  1. The authors did not mention why the hippocampus was investigated in this study. How is this brain area associated with Meth? Please include in the introduction section and provide the references.
  2. The possible factors and mechanisms that regulate CD200/CD200r should be mentioned.
  3. Line 65: “The communication between these two types of cells …..” What kind of cells the authors would like to mention in this sentence?

Materials and methods

  1. Why do the primary neurons were prepared from the hippocampus, whereas the primary astrocytes and microglia were prepared from the cortex and hippocampus?
  2. Why do the authors use 100 mM Meth in this study? Please provide an explanation and references.
  3. References in the methods section are needed for: Immunocytochemistry, Phagocytic assay, RNA study, and Flow cytometry.

Results/ Figures

  1. The sentence “Symbol colors represent the mean of each independent cell culture and the violin plots …..” and the statistical analysis were stated too many times in the same figure legend. Please mention only one time at the end of each paragraph.
  2. Line 362: the authors suggested that Meth induced microglia activation via astrocytes is attenuated by neuronal action. Please provide more explanation since the mRNA expression of both arginase 1 and iNOS were increased while both neurons and microglia were exposed to ACM Meth.
  3. Please state why the VGlut1 was observed and used for the co-localization with the PSD95. Please also mention, probably in the Discussion section, what are the difference in the interpretation of the puncta number, puncta area, and the puncta intensity results.

Discussion

  1. “CD200r” or “CD200R”? please use the same abbreviation.
  2. Line 535: the sentence “Although the exact mechanisms regulating ….” Is too long. Please recheck if punctuation marks were missing.

Reviewer 2 Report

The current report investigates the link between methamphetamine microglia using in vitro systems. There are a number of major and minor concerns diminishing enthusiasm; these concerns are detailed below.

Major concerns:

  • The rationale/justification for using hippocampal cultures is limited. The primary pharmacological targets of meth, i.e. monoamine transporters and VMAT2, are not expressed with the exception of potential astrocytic DAT expression. Even with astrocytic DAT expression, there is no monoamine input so meth does not have the capacity to increase extracellular monoamines in the in vitro systems used throughout the study. Not only is this significant issue not addressed but there is no data or even discussion relating to what meth might be binding to that could explain observed effects. Secondary to this is the rather high concentration of meth used which also lacks rationale/justification. Together the in vitro systems used and high concentration raise questions of translational relevance. This point likely underlies the large amount of negative data; had these studies been done with cultured dopamine neurons or co-cultures so that the hippocampal neurons had dopaminergic input there would likely be very different results. 
  • In 3.2 it is stated that microglial activation is attenuated by neuronal action but there is no data indicative of neuronal activity or function nor is there any discussion (or data) relating to mechanism. The associative changes reported are difficult to interpret when it is unclear what meth is targeting in this system and this is not discussed. Also in section 3.4 authors report higher levels of active synapses; evidence for this is indirect and experiments assessing synaptic function are lacking.
  • In the discussion authors state that “The primary goal of this study was to further contribute to clarify the mechanisms involved in meth-induced neurotoxicity . . . ‘; there are no experiments in the current study that directly examine toxicity or test for relationships between current findings and toxicity.

Minor Concerns:

  • Both the introduction and discussion are limited in scope and lacking with respect to incorporating existing meth tox literature
  • It is unclear why iba1 quantification is omitted for experiments in figures 2 and 3 whereas arginase is absent in figure 1
  • Text in the abstract and introduction discuss meth toxicity but then also emphasize emerging roles of astrocytes in addiction; this is distracting given that the authors’ previous paper, from which the current work seems to be an extension of, is within the realm of toxicity, particularly with the high concentration of 100µM used.
  • In the abstract it is stated ‘We observed also increased plasticity’; this is vague and would benefit from more clear phrasing in the abstract.
  • In the intro it is made clear that previous findings were using an acute meth binge but in the first sentence of the discussion findings are referred to with ‘acute meth exposure’; acute meth such as a single one day injection versus a binge paradigm should be clearly distinguished

Round 2

Reviewer 2 Report

In the methods it is stated that striatal neurons were used but in the text it states striatal GABAergic interneurons (L293) and striatal interneurons (L325). Are these cultured neurons actual interneurons or are they spiny projection neurons or perhaps an unidentified mix of striatal neurons?